# Multimodal, Technology-Assisted Intervention for the Management of Menopause after Cancer Improves Cancer-Related Quality of Life—Results from the Menopause after Cancer (Mac) Study

**DOI:** 10.3390/cancers16061127

**Published:** 2024-03-12

**Authors:** Fionán Donohoe, Yvonne O’Meara, Aidin Roberts, Louise Comerford, Ivaila Valcheva, Una Kearns, Marie Galligan, Michaela J. Higgins, Alasdair L. Henry, Catherine M. Kelly, Janice M. Walshe, Martha Hickey, Donal J. Brennan

**Affiliations:** 1UCD Gynaecological Oncology Group, UCD School of Medicine, Mater Misericordiae University Hospital, Eccles Street, D07 AX57 Dublin, Ireland; fionandonohoe@gmail.com (F.D.);; 2myPatientSpace Ltd., K36 A022 Dublin, Ireland; 3Clinical Research Centre, UCD School of Medicine, Mater Misericordiae University Hospital, Eccles Street, D07 AX57 Dublin, Ireland; 4Department of Medical Oncology, St. Vincent’s University Hospital, Elm Park, D04 T6F4 Dublin, Ireland; 5Big Health Ltd., London WC1H 9LT, UK; 6Big Health Ltd., San Francisco, CA 94108, USA; 7Department of Medical Oncology, Mater Misericordiae University Hospital, Eccles Street, D07 AX57 Dublin, Ireland; 8Department of Obstetrics and Gynaecology, University of Melbourne and The Royal Women’s Hospital, Parkville, VIC 3052, Australia; 9Systems Biology Ireland, UCD School of Medicine, Belfield, D04 V1W8 Dublin, Ireland

**Keywords:** menopause, survivorship, quality of life, insomnia

## Abstract

**Simple Summary:**

Cancer and cancer treatment can induce menopause symptoms such as hot flushes, night sweats and poor sleep. Menopausal hormone therapy (MHT) is the most effective treatment for these symptoms but may be contraindicated following certain cancer diagnoses. This study assesses if a composite intervention aimed to target these symptoms in women with a history of cancer and a contraindication to standard MHT could improve their quality of life over six months. We found a combination of non-hormonal medications to address daytime hot flushes and/or night sweats with digital cognitive behavioral therapy for insomnia, the provision of self-management strategies and identification of a partner or other support person resulting in clinically relevant and statistically significant improvements in cancer-specific quality of life, insomnia symptoms, the frequency of hot flushes and night sweats and the degree of bother of these symptoms on day to day functioning.

**Abstract:**

Background: Vasomotor symptoms (VMSs) associated with menopause represent a significant challenge for many patients after cancer treatment, particularly if conventional menopausal hormone therapy (MHT) is contraindicated. Methods: The Menopause after Cancer (MAC) Study (NCT04766229) was a single-arm phase II trial examining the impact of a composite intervention consisting of (1) the use of non-hormonal pharmacotherapy to manage VMS, (2) digital cognitive behavioral therapy for insomnia (dCBT-I) using Sleepio (Big Health), (3) self-management strategies for VMS delivered via the myPatientSpace mobile application and (4) nomination of an additional support person/partner on quality of life (QoL) in women with moderate-to-severe VMS after cancer. The primary outcome was a change in cancer-specific global QoL assessed by the EORTC QLC C-30 v3 at 6 months. Secondary outcomes included the frequency of VMS, the bother/interference of VMS and insomnia symptoms. Results: In total, 204 women (82% previous breast cancer) with a median age of 49 years (range 28–66) were recruited. A total of 120 women completed the protocol. Global QoL scores increased from 62.2 (95%CI 58.6–65.4) to 70.4 (95%CI 67.1–73.8) at 6 months (*p* < 0.001) in the intention to treatment (ITT) cohort (*n* = 204) and from 62 (95%CI 58.6–65.4) to 70.4 (95%CI 67.1–73.8) at 6 months (*p* < 0.001) in the per-protocol (PP) cohort (*n* = 120). At least 50% reductions were noticed in the frequency of VMS as well as the degree of bother/interference of VMS at six months. The prevalence of insomnia reduced from 93.1% at the baseline to 45.2% at 6 months (*p* < 0.001). The Sleep Condition Indicator increased from 8.5 (SEM 0.4) to 17.3 (SEM 0.5) (*p* < 0.0005) in the ITT cohort and 7.9 (SEM 0.4) to 17.3 (SEM 0.5) (*p* < 0.001) in the PP cohort. Conclusions: A targeted composite intervention improves the quality of life for cancer patients with frequent and bothersome vasomotor symptoms with additional benefits on frequency, the bother/interference of VMS and insomnia symptoms.

## 1. Introduction

Menopausal symptoms are common after cancer treatment and may be severe. Cancer-related menopausal symptoms may be induced or exacerbated in several ways, including oophorectomy, pelvic radiotherapy, cytotoxic chemotherapy and anti-endocrine therapy. Those diagnosed with estrogen-sensitive cancers are commonly advised to discontinue menopausal hormone therapy (MHT), which may provoke resurgent symptoms. Breast and gynecologic cancers and their treatments are commonly complicated by menopause symptoms, but non-reproductive tract cancers can also be implicated, particularly hematological malignancies, as stem cell transplantation almost always results in premature menopause [1]. In addition, with the rising incidence of early-onset colorectal cancers worldwide [2], cancer-associated menopause is also becoming more prevalent in this population.

MHT is the most effective treatment for VMS following natural menopause [3], reducing symptoms by ~85% and may improve sleep and quality of life [4]. MHT may be contraindicated following cancer, particularly estrogen-sensitive breast cancer [5,6,7]. Growing evidence supports the efficacy of non-hormonal pharmacotherapies for VMS, but this evidence is derived from studies in breast cancer [8]. Reductions up to 50–60% in the frequency of VMS have been demonstrated using the selective serotonin noradrenaline uptake inhibitor (SNRI) venlafaxine [9,10,11,12,13,14], serotonin reuptake inhibitors (SSRIs) such as citalopram [15], sertraline [16,17], fluoxetine [18] and duloxetine [19], the anti-cholinergic oxybutynin [20] and the anticonvulsants gabapentin [21,22] and pregabalin [23]. The use of these non-hormonal has been recommended by multiple national bodies and societies for managing VMS in breast cancer survivors, although their use is ‘off-label’ [24,25,26,27].

Sleep disturbance is commonly experienced by cancer survivors, with rates ranging from 25 to 59% [28]. Indeed, cancer-associated insomnia has been linked to poorer quality of life [29,30], memory problems [31], reduced survival [32] and may even promote tumor growth [33,34]. Cognitive behavioral therapy for insomnia (CBT-I) is the gold standard, guideline-recommended first-line treatment for insomnia [35,36,37]. It comprises evidence-based cognitive and behavioral interventions to address insomnia symptoms, including sleep restriction, stimulus control and cognitive therapy [38]. In the context of cancer, meta-analyses demonstrate the effectiveness of CBT-I at reducing key insomnia outcomes when delivered in a variety of formats, including on a one-to-one basis, self-guided and digitally [39]. It has also been shown to be the most effective intervention for insomnia symptoms in women with VMS [40]. Recently, a specific digital CBT treatment for insomnia (Sleepio) was endorsed by NICE as the recommended treatment for insomnia in the community [41].

Given the close links between cancer-related menopause and insomnia, we conducted a prospective, single-arm phase II trial to evaluate a multimodal intervention delivering a combination of non-hormonal pharmacotherapy and dCBT-I in women with bothersome VMS after cancer treatment. The primary outcome of the Menopause After Cancer (MAC) trial was the impact on cancer-specific global quality of life over a six-month period measured using the EORTC-QLQ-C30 [42], with secondary outcomes of the frequency of VMS and bother/interference of VMS, both measured by the Hot Flush Rating Scale (HFRS) [43] and degree of insomnia symptoms measured by the Sleep Condition Indicator (SCI) [44].

## 2. Materials and Methods

The MAC study consisted of a composite intervention (Figure 1A) with the following four parts: (1) non-hormonal pharmacotherapy to manage VMS, (2) dCBT-I using Sleepio (Big Health Ltd., London, UK; San Francisco, CA, USA.) [45], (3) self-management strategies for VMS delivered via a mobile application called myPatientSpace and (4) the nomination of an additional support person/partner as outlined in the previously published trial protocol [46]. Additional descriptions of dCBT-I, myPatientSpace and support partners are provided in Appendix A. Potentially eligible participants were identified through routine clinical care, reviewed via video calling software (Zoom, version 5, San Jose, CA, USA) and consented virtually to participate in the study. All outcome data were collected through the myPatientSpace (Dublin, Ireland) app and accessed by the study team centrally.

The inclusion criteria were as follows:Female aged 18 or over;At least five VMS episodes in an average 24 h period;Moderate degree of bother from these symptoms defined as a score of 5.3 or higher on HFRS [43];A prior or current history of cancer;A contraindication to standard MHT;Confident use of a smartphone.

The exclusion criteria were as follows:Eastern Cooperative Oncology Group (ECOG) performance status of 3 or higher;Use of study medications to manage VMS in the preceding six months;Use of CBT-I in the preceding six months;Any contraindication to study medications;Limited spoken or written English;No internet access or not confident with smartphone use.

### 2.1. Non-Hormonal Pharmacotherapy

For predominantly daytime symptoms, patients were prescribed SSRI (citalopram) or SNRI (venlafaxine) based on the physician’s preference (Figure 1B). Citalopram, starting at a dose of 10 mg, was increased to a maximum of 30 mg if required [15,47,48,49]. A dose increase to 20 mg was considered after 4 weeks based on clinical response [15]. Venlafaxine commenced at a dose of 37.5 mg and increased to 75 mg after four weeks, depending on response and tolerance. If symptoms were most bothersome at night, patients were prescribed gabapentin starting at 300 mg one hour before bedtime, increasing by 100 mg every three nights until sleep disturbance or the onset of side effects was reduced, or a maximum of 900 mg was reached [50]. If symptoms were equally bothersome during the day and at night-time, the medications were used in combination. In this situation, patients were commenced on 10 mg of citalopram in the morning and gabapentin at 300 mg at bedtime for three days. The gabapentin dose could then be increased to twice daily for three more days if required and tolerated, and then after a further three days, to three times a day, again, if tolerated and required [21].

### 2.2. Outcomes

The primary outcome of this study was the six-month change in cancer-specific global health status/quality of life as assessed by the EORTC-QLQ-C30 version 3 [42] relative to the baseline. The EORTC-QLQ-C30 was completed at the baseline, four weeks, three months and, finally, six months post the commencement of the intervention. Participants received push notifications through the myPatientSpace app when the outcome measures were due to be completed. Secondary outcomes included the frequency of both night sweats and daytime hot flashes as well as the degree of symptom bother. These were measured using the HFRS instrument [43] at each study timepoint. This was a self-report measure of the frequency and problem-rating of vasomotor symptoms over the preceding week [51]. Secondary outcomes also included insomnia symptoms, measured with the 8-item Sleep Condition Indicator (SCI) [44]. SCI scores of 16 or lower were indicative of insomnia disorder, and a change in SCI score of 7 points or more was considered a reliable change [52].

### 2.3. Sample Size Calculation

The EORTC defines the smallest clinically relevant change for the global health status/quality of life scale as 5 points [53]. We anticipated that most participants would have previous breast cancer, so baseline global health status scores were extracted from an EORTC registry of trials and Project Data Sphere (PDS) [54,55]. This demonstrated that the baseline global quality of life in early breast cancer was 76.9 (SD 19.20) in the EORTC cohort and 72.4 (SD-18.8) in the PDS cohort. In metastatic breast cancer, the global quality of life scores were 57.7 (SD-23.1) in the EORTC cohort and 54.6 (SD-20.1) in the PDS cohort. As we aimed to recruit women with both early and metastatic cancer, we estimated that the mean pre-treatment EORTC-QLQ-C30 global health status/quality of life score in this cohort of women would be 65 (SD 20). To detect a 5-point improvement with 90% power and a two-sided 5% significance level required a minimum of 171 patients. To account for a 15% drop-out rate, we aimed to recruit 201 participants.

### 2.4. Statistical Analysis

Statistical analysis was performed using IBM SPSS statistical package version 27 (IBM, Chicago, IL, USA) and R version 4.3. All outcome variables were expressed as means with 95% confidence intervals (95% CIs). For all outcomes, the results for the intention to treat (ITT) and the per-protocol (PP) populations were reported separately. The ITT population is the entire cohort, while the PP population includes only those who completed the final six-month outcome measure for the primary endpoint. Means were compared at the baseline and at six months using a *t*-test as data were normally distributed. Categorical data were compared using a chi-squared test. *p* values of less than 0.05 were considered statistically significant.

Exploratory analysis was conducted to explore the association of baseline factors with improved quality of life. Subjects with available baseline scores were categorized as having an improvement in quality of life (an increase of 5 points at 6 months relative to the baseline) or not. Where QoL scores were missing, the most recent score was carried forward. Tests of association between baseline characteristics and improved quality of life were then evaluated using Fisher’s Exact test for categorical baseline characteristics and Mann–Whitney U tests or Kruskal–Wallis’ ANOVA test for numeric baseline characteristics.

### 2.5. Ethics

Ethical approval was granted from the research ethics committee of St. Vincent’s Healthcare group, Elm Park, Dublin 4, Ireland (RS21-002), and the study was prospectively registered on clinicaltrials.gov with reference number NCT04766229.

## 3. Results

In total, 356 women were referred for consideration in this study. A total of 102 declined to participate, and 49 did not meet the inclusion criteria. In total, 205 participants were recruited from June 2021 until May 2022; 1 participant withdrew consent after enrolment. Data are available on 204 women for analysis (Figure A1). Total recruitment occurred over 47 weeks, and baseline demographic information for the ITT and PP populations is presented in Table 1. The median age was 49 (range 28–66); 99% were Caucasian, and over 85% of patients had a tertiary education. Eighty-two percent of patients had a history of breast cancer; 70% of these were taking endocrine therapy, and 93% were disease-free at the time of enrolment. Baseline features were similar across both ITT and PP populations.

The baseline mean number of hot flashes was 11.8 per day (range 0–60) and 5.1 (range 0–30). night sweats. The inclusion criteria required participants to experience at least five daytime flashes and/or night sweats; some complained of at least five nocturnal VMS with no daytime symptoms, and others experienced at least five daytime symptoms and zero nocturnal symptoms, explaining why the baseline frequency of VMS ranges from zero. Forty-eight percent (98/204) of participants experienced VMS predominantly at night, 31% (64/204) reported nocturnal and daytime symptoms of equal frequency and 21% (42/204) reported VMS predominantly during the day.

Regarding menopause etiology, 42% (85/204) of participants reported chemotherapy-induced VMS, 17% (35/204) experienced surgical menopause, 29% (59/204) attributed their symptoms to endocrine therapy with or without ovarian suppression, 8% (16/204) had voluntarily stopped MHT after a cancer diagnosis and 3% (7/204) had a remote history of cancer but developed problematic VMS over the natural menopause transition. In total, 198 participants (97%) were ineligible for MHT for oncological reasons. The contraindications for the remaining six included cardiovascular comorbidities, prior venous thromboembolism and personal choice.

### 3.1. Compliance

A total of 93% of participants (189/204) completed the baseline EORTC-QLQ-C30 measurement with 162 (79.4%) at 4 weeks, falling to 120 (58.8%) participants completing the questionnaire at the six-month timepoint. (Figure 2A). Thus, the dropout rate was 41%.

The nonhormonal pharmacotherapy regimens used are outlined in Figure 2B. Gabapentin in a single night-time dose was prescribed for 51% (104/204) of participants, while 20.1% (41/204) prescribed both citalopram and gabapentin for day and nighttime symptoms, 16.7% (34/204) were prescribed citalopram, and one patient was prescribed venlafaxine due to a potential medication interaction between citalopram and plaquenil. In total, 11.8% (24/204) were prescribed gabapentin in divided daily doses for daytime predominant symptoms in the setting of prior SSRI or SNRI use for depression or anxiety. The overall compliance rate with prescribed medications, based on participant self-report, was 64.2% (131/204) at three months and 49.5% (101/204) at six months. The highest compliance rate was seen in combined treatment with citalopram and gabapentin (Figure 2C), suggesting women with worse symptoms at the baseline were most likely to remain on medication. Compliance with dCBTi was recorded by the Sleepio platform. In total, 170 (83.3%) participants logged in and created a Sleepio account; 161 (94.7%) of these commenced the first CBT session, with 62 (36.5%) completing all sessions (Figure 2D).

### 3.2. The Impact of the Composite MAC Intervention on Global Health Status

In the ITT cohort, the baseline mean EORTC-QLQ-C30 score increased from 62.2, 95% CI [58.6–65.4] to 70.4, 95% CI [67.1–73.8] at 6 months (*p* < 0.001). In the PP cohort, the mean score increased from 62.0, 95% CI [58.6–65.4] at the baseline to 70.4, 95% CI [67.1–73.8] at 6 months (*p* < 0.001) (Figure 3A). Despite a higher-than-anticipated dropout rate, we detected an 8.2-point improvement in global health status in the ITT cohort and an 8.4-point increase in the PP cohort. This represents a medium-sized clinical improvement in global health status as defined by the EORTC [42].

### 3.3. Baseline Quality of Life Was the Main Predictor of Benefit from the MAC Intervention

Based on our previous observation that compliance with non-hormonal pharmacotherapy was highest in women who were prescribed both citalopram and gabapentin, we sought to assess if those with worse symptoms at the baseline and consequently lower quality of life would gain the greatest benefit from MAC intervention. Patients were categorized into three approximately equally sized groups depending on their global health status scores at the baseline and were categorized as having ‘low’ (<50th centile), mid (50th—75th centile) and high (<75th centile) global health status. Boxplots of subject global health status scores defined by these thresholds were constructed, which demonstrated the greatest changes were seen in those who started out with lower global health status (Figure 3B). Further exploratory analysis was conducted to identify associations between observed baseline characteristics, the rates of completion of intervention elements, and the incidence of clinically meaningful improvement in global health status from the baseline to six months (Table 2). A baseline global health status score was the only baseline variable strongly associated with a clinically significant improvement in global health status over 6 months of the intervention (*p* < 0.001). Improvers were also marginally more likely than non-improvers to have a breast cancer diagnosis as opposed to other cancers, though this association was not statistically significant (*p* = 0.052). Other baseline variables were not associated with a clinically significant improvement in a global health status score over 6 months. Clinically significant improvements in global health status were also associated with the continuation of medications at six months (*p* < 0.001) and the completion of Sleepio (*p* = 0.008) (Table 2), suggesting that those who continued to engage with these parts of the intervention were more likely to see an improvement, although these data are biased by loss to follow up.

### 3.4. Menopause Outcomes

In the ITT cohort, the mean frequency of daytime VMS fell from 11.8, 95% CI [10.45–13.10] flushes per day at the baseline to 6.0, 95% CI [4.46–7.52] at six months (*p* < 0.001). The frequency of night-time VMS fell from 5.1, 95% CI [4.41–5.82] to 1.5, 95% CI [1.14–1.84] sweats per night at six months (*p* < 0.001). For the PP cohort, the mean daytime frequency fell from 12.7, 95% CI [10.91–14.56] flushes per day at the baseline to 6.0, 95% CI [4.49–7.59] flushes per day at six months (*p* < 0.001) For night-time symptoms in the PP cohort, the mean frequency fell from 5.2, 95% CI [4.40–6.05] sweats per night at the baseline to 1.5, 95% CI [1.15–1.86] sweats per night at 6 months (*p* < 0.001). This represented a fifty percent decrease in the frequency of daily hot flushes and a seventy percent decrease in night sweats, with the greatest effect seen in the first 4 weeks of the study (Figure 4A,B). The frequency of all VMSs (day and night times) was reduced by approximately 58% for both the ITT (58% reduction) and PP populations (Figure 4C).

HFRS scores demonstrated lessening of the bother/interference of VMS over the trial period. All 204 participants completed baseline HFRS data as these were completed as part of screening for inclusion. A total of 167, 137 and 121 participants completed the questionnaire at the 4-week, 3-month and 6-month timepoints, respectively. The HFRS is scored from 1 to 10, with higher scores indicating a greater degree of bother/interference.

In the ITT cohort, the mean HFRS decreased from 7.6, 95% CI [7.37–7.74] at the baseline to 3.4, 95% CI [2.98–3.75] at six months (*p* < 0.001). For the PP cohort, the mean HFRS decreased from 7.5, 95% CI [7.30–7.77] at the baseline to 3.4, 95% CI [2.96–3.74] at 6 months (*p* = 0.036) (Figure 4D). In clinical terms, this represents a 55% decrease in the bother/interference of VMS. As with the improvements seen with the frequency of VMS, the greatest improvements in the bother/interference of VMS were seen at 4 weeks, with a relative plateauing of the effect over subsequent time (Figure 4D).

### 3.5. Sleep Outcomes

The principal sleep outcome was insomnia symptoms assessed by the SCI. In total, 188 (92.2%) participants completed the baseline SCI assessment, 129 (63.2%) the three-month assessment and 124 (60.8%) completed the final SCI questionnaire.

In the ITT cohort, mean SCI scores increased from 8.49 (SEM 0.4) at the baseline to 17.26 (SEM 0.5) at 6 months (*p* < 0.0005) (7.86 (SEM 0.4) to 17.24 (SEM 0.5) (*p* < 0.001) in the PP cohort), with most of the benefits seen at three months (Figure 4E). Although not an inclusion criterion for this study, 93.1% (175/188) of participants met the criteria for insomnia disorder (SCI ≤ 16) at the baseline, with 45.2% (56/124) meeting the same criteria at six months (*p* < 0.00001). Further subset analysis was performed on SCI scores from participants who completed all six sessions of Sleepio (*n* = 62), those who undertook at least one session of Sleepio (*n* = 161), and those who did not undertake any sessions of Sleepio (*n* = 43). Gabapentin used for VMS, as in this study, has also been shown to be effective at improving sleep [22]; thus, it is difficult to ascertain if any improvements in sleep were mediated by pharmacotherapy, dCBT-I or both. Therefore, the rates of gabapentin prescription and compliance in each of these sub-cohorts are also reported.

For Sleepio completers, the baseline SCI scores were 7.48 (SEM 0.4). Improvements were similar to the ITT and PP cohorts, with scores of 16.17 (SEM 0.8) at the three-month timepoint and 17.43 (SEM 0.8) at the six-month timepoint. In total, 50/62 (80.6%) of these participants were prescribed gabapentin in some dosing regimen at the study’s outset. At the six-month timepoint, 41 participants were still taking gabapentin, which represents 83.7% (41/49) of all participants who completed the SCI measure at the six-month timepoint from this sub-cohort.

In those who accessed at least one session of Sleepio, SCI scores were similar to those who completed all six sessions. Baseline SCI scores were 8.31 (SEM 0.4). The three-month scores also improved to 16.83 (SEM 0.5), and the six-month scores remained similar at 17.12 (SEM 0.5). In total, 134 of these participants (83.2%) were prescribed gabapentin at the study’s outset. In total, 74 were still taking gabapentin at the six-month timepoint, which represents 67.3% (74/110) of participants in this sub-cohort who completed the SCI measure at the six-month timepoint.

Interestingly, the small number (*n* = 43) of participants who elected not to use dCBT-I still saw improvements in sleep. Their baseline SCI scores were slightly higher than other cohorts at 9.48 (SEM 1.2). The three-month mean scores were 17.81 (SEM 1.5), with six-month mean SCI scores increasing to 18.4 (SEM 1.6). Here, 35 (81.4%) were prescribed gabapentin at the study’s outset. By the six-month timepoint, only 11 remained taking gabapentin, which represents 78.6% (11/14) of participants in this sub-cohort who completed the SCI measure at the six-month timepoint. This information is summarized in Table 3. Given these relatively high rates of gabapentin prescription, it is difficult to ascertain whether the improvements in sleep were due to the Sleepio content or the use of gabapentin.

## 4. Discussion

The MAC study met its primary endpoint and resulted in a clinically meaningful and statistically significant improvement in the global health status scale of the EORTC-QLQ-C30 at six months, with most benefits seen within four weeks of the study’s initiation. We elected to use the EORTC-QLQ-C30 as the primary outcome measure. This measures cancer-specific quality of life rather than menopause-specific quality of life. This instrument was chosen as it was felt that cancer-specific quality of life was a more widely recognized and identifiable parameter to allow wider communication of the research findings. In addition, an approximate 50% reduction in the frequency of VMS and clinically meaningful improvements in sleep were observed. Given the single-arm design of this study and the composite nature of the intervention, it is difficult to say what elements of the intervention are responsible for these improvements. This single-arm design was chosen partly to give participants the greatest chance of experiencing improvements but also because the composite nature of the intervention effect sizes was difficult to estimate. Now that this study is complete, future research could include the randomization of participants to certain parts of the intervention; for example, participants could be randomized to include Sleepio with the other aspects of the intervention to determine which aspects may be having the greatest impact on the outcome. Including participants’ perceptions of what were the most beneficial aspects of the intervention through end-of-study surveys could also have allowed some insight into what elements of the intervention were most effective for managing these symptoms. Again, this could be included in future research.

The rate of loss to follow up was greater than expected and likely driven by the duration of follow up. Previous studies of SSRIs and SNRIs for VMS show discontinuation rates of between 8.5 and 41%, with lower discontinuation rates seen in studies with a shorter duration of follow up. For previous studies on gabapentin, discontinuation rates are approximately 25% [21,22]. The longest duration of follow up for any of these studies is 12 weeks. Studies with a shorter follow up are associated with lower discontinuation rates. While this study seems to have a high dropout rate of 41%, this is not unexpected given the long follow up period. Future randomized studies are needed to confirm the effect sizes seen in this study, especially as very few other studies on non-hormonal agents for VMS have examined impacts on quality of life [16,17,20,56]. Interestingly, in these studies, no changes in quality of life were seen. Data on the specific reasons for individual participant attrition in this study were not collected, so beyond the association with the duration of follow up discussed above, it is difficult to say what else might have contributed to this. A commonly cited cause for attrition is this study’s burden on participants for issues such as traveling for appointments and follow up [57,58,59]. To reduce this burden, the digital collection of outcome measures through the myPatientSpace app, a virtual consent process and phone call follow up visits were employed. While access to Sleepio was provided without charge, participants were required to self-fund the medications, which might have been a further reason for attrition.

This is also the first study to assess the impact of Sleepio in cancer patients with troublesome VMS and is the first to measure the effect of Sleepio on cancer-related quality of life. Sleep-related quality of life has been reported in one prior study using Sleepio, with significant improvements observed, but this was not in a cancer-specific population [60]. Compliance rates with Sleepio were lower in this study than in previous studies on Sleepio. This is probably because insomnia was not an inclusion criterion for this study, and many participants might not have felt that the content was relevant to them due to their sleep problems not being severe. This has been reported in other studies on Sleepio that did not have a specific severity of sleep disturbance required for inclusion [61]. Insomnia was, nevertheless, common in this population, even in those who did not access or complete Sleepio. Randomized trials are needed to inform the impact of Sleepio in female cancer survivors [62].

As the largest improvements in quality of life were seen in those with lower baseline scores, future studies could consider using a certain threshold of the quality-of-life score as an inclusion factor to try and identify those who could benefit the most. In addition, the largest improvement in quality-of-life scores in this study was seen at four weeks and remained relatively stable throughout the rest of the study. In future studies, using a shorter interval of follow up might decrease the drop-out rate, provide more robust data and possibly demonstrate that shorter interventions are effective.

Medication compliance was poorest in those prescribed gabapentin in divided daily doses, which might be due to the sedating side effects of the drug. This was greater than previously published studies which only followed patients for between 8 and 12 weeks after the initiation of therapy and, thus, might not have identified reduced compliance over longer follow up periods [22,63]. In fact, given the relatively short durations of follow up seen in most of the trials of these non-hormonal agents, little is known about the longer-term efficacy of using non-hormonal agents to control VMS.

Side effects of anti-endocrine therapy for breast cancer (which include VMS and insomnia) are known to affect medication adherence, which can negatively impact oncological outcomes [64,65]. Proactively managing these side effects is an important aspect of follow up and survivorship care, which can have important benefits from a quality-of-life standpoint, as shown in this study, but also from a disease management perspective.

## 5. Conclusions

In conclusion, menopausal symptoms are common after cancer and may impact on quality of life. With improving treatments and increasing life expectancy, survival rates from cancer are likely to increase, which forces us to look more closely at the longer-term effects of cancer and cancer treatment and how best we can help address these challenges as they become more commonly encountered in clinical practice. This study demonstrates that it is possible to conduct this kind of research and that improvements in cancer-specific quality of life may be seen from as early as four weeks following the commencement of an intervention such as this and be maintained for up to six months.

## Figures and Tables

**Figure 1 cancers-16-01127-f001:**
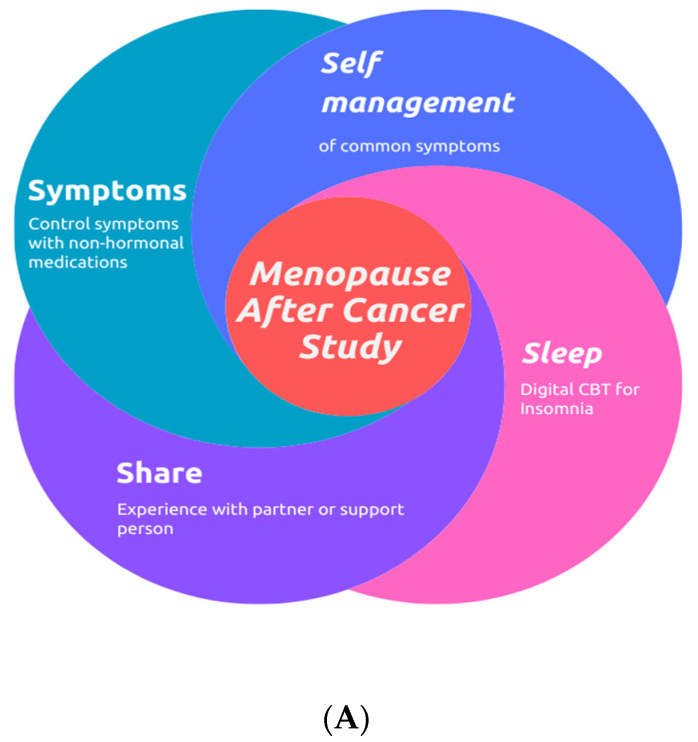
(**A**): Schematic representing the composite intervention of the MAC study. (**B**): Flowchart demonstrating specifics of the composite intervention in the MAC study.

**Figure 2 cancers-16-01127-f002:**
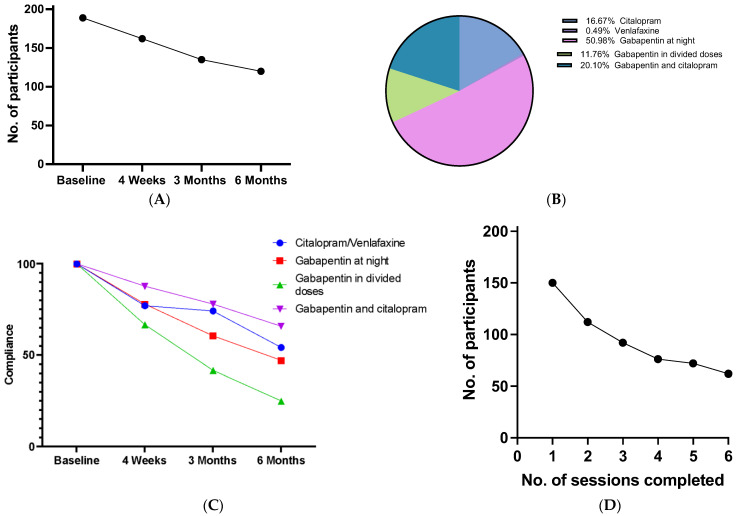
Compliance with MAC intervention. (**A**): Graph demonstrating the number of participants who completed the EORTC-QLQ-C30 questionnaire at each timepoint. (**B**): Chart demonstrating how often each regimen was prescribed in the study. (**C**): Graph demonstrating compliance with each drug regimen over the course of the study. (**D**): Graph demonstrating the number of participants who completed each of the six sessions of dCBT-I in Sleepio.

**Figure 3 cancers-16-01127-f003:**
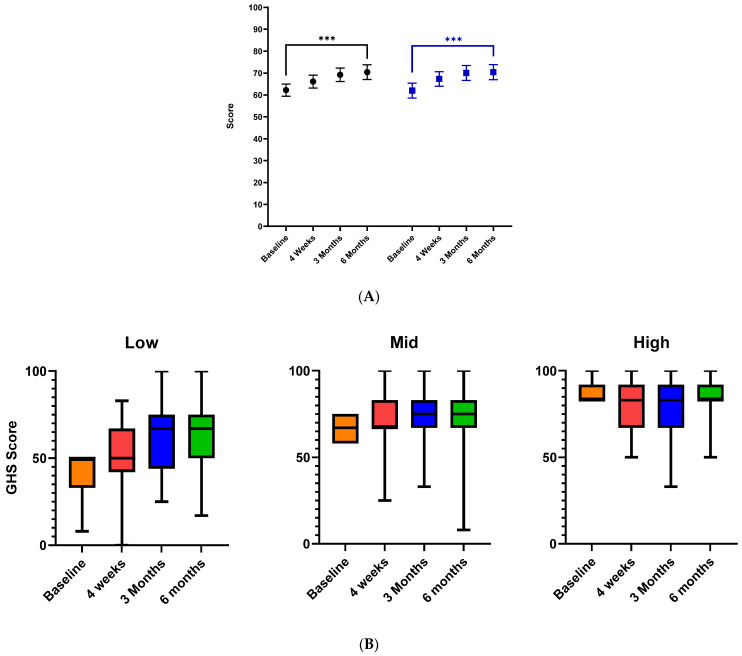
Changes in all scales of the EORTC-QLQ-C30 in the ITT and PP cohorts. (**A**): Mean and 95% CI in the global health status scale in the EORTC-QLQ-C30 instrument for the ITT cohort (shown in black) and the PP cohort (shown in blue). (**B**): Box plot showing global health status scores categorized according to low, mid or high global health status scores at the baseline. Those with the lowest scores at baseline saw the greatest improvement in these scores. *** denotes statistical significance < 0.005.

**Figure 4 cancers-16-01127-f004:**
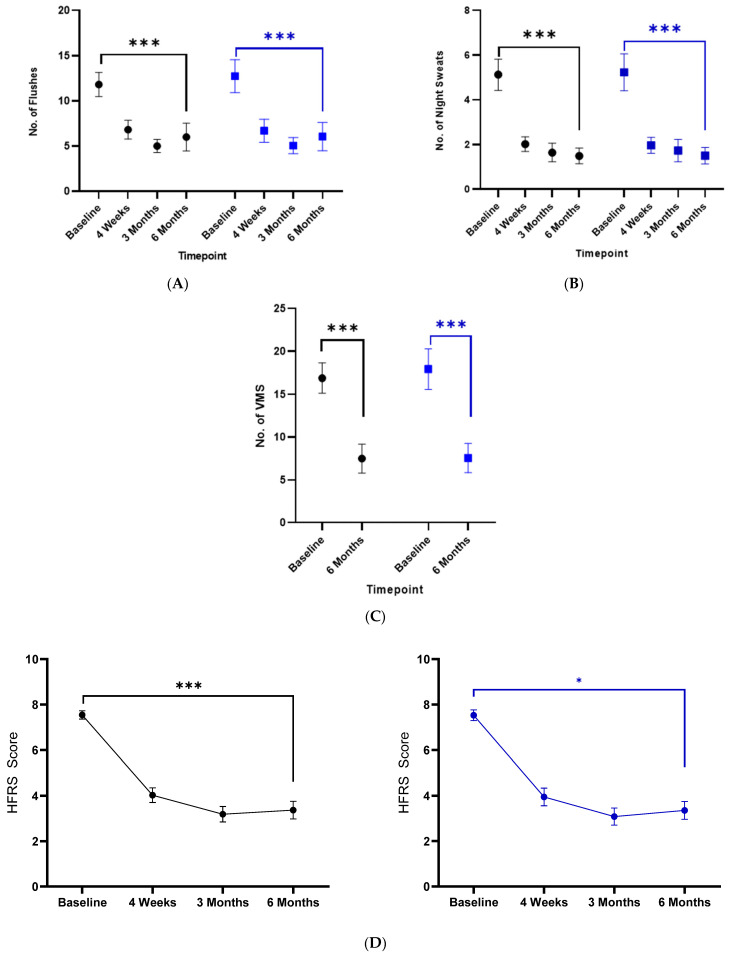
Changes in menopause and sleep outcomes ITT and PP cohorts. (**A**): Mean and 95% CI for the frequency of daytime hot flashes for the ITT cohort (shown in black) and the PP cohort (shown in blue) over the study period. (**B**): Mean and 95% CI for the frequency of night sweats for the ITT cohort (shown in black) and the PP cohort (shown in blue) over the study period. (**C**): Mean and 95% CI frequency for all VMSs for the ITT cohort (shown in black) and the PP cohort (shown in blue) over the study period. (**D**): Mean and 95% CI for Hot Flush Rating Scale scores in the ITT cohort (shown in black) and the PP cohort (shown in blue). (**E**): Mean and SEM for the Sleep Condition Indicator in the ITT cohort (shown in black) and the PP cohort (shown in blue). * denotes statistical significance <0.05, *** denotes statistical significance <0.005.

**Table 1 cancers-16-01127-t001:** Baseline demographics (figures in parentheses represent percentages unless otherwise stated).

		ITT (*n* = 204)	PP (*n* = 120)
Median Age (Range, IQR)		49 (28–66, 45–53)	50 (28–66, 46–53)
Ethnicity	White	202 (99)	118 (98.3)
	Asian/Asian Irish	2 (1)	2 (1.7)
Educational level	Primary	1 (0.5)	1 (0.8)
	Secondary	29 (14.2)	16 (13.3)
	Third level	138 (67.6)	87 (72.5)
	Third level > 4 years	36 (17.6)	16 (13.3)
Smoking status	Current smoker	12 (5.9)	7 (5.8)
	Ex smoker	71 (34.8)	42 (35)
	Non-smoker	121 (59.3)	71 (59.2)
Alcohol consumption	Non-drinkers	52 (25.4)	33 (27.5)
	1–5 units/week	109 (53.4)	65 (54.2)
	6–10 units/week	36 (17.6)	19 (15.8)
	>10 units/week	7 (3.4)	3 (2.5)
Exercise frequency (active for >30 min)	5–7 times/week	112 (54.9)	72 (60)
	1–5 times/week	69 (33.8)	35 (29.2)
	Stopped	21 (10.3)	12 (10.8)
	Never	2 (1)	0
Diagnosis	Breast cancer	167 (81.9)	102 (85)
	Ovarian cancer	18 (8.8)	7 (5.8)
	Endometrial cancer	12 (5.9)	7 (5.8)
	Other	7 (3.4)	4 (3.3)
Stage at diagnosis	Stage 1	58 (28.4)	32 (26.7)
	Stage 2	70 (34.3)	41 (34.2)
	Stage 3	44 (21.6)	25 (20.8)
	Stage 4	11 (5.4)	7 (5.8)
	Stage unknown	21 (10.3)	15 (12.5)
Treatment—Breast cancer	Surgery alone	42 (25.1)	30 (29.4)
ITT (*n* = 167)	Surgery + chemotherapy	21 (12.6)	10 (9.8)
PP (*n* = 102)	Surgery + chemotherapy + radiotherapy	102 (61.1)	60 (58.8)
	Chemotherapy alone		
		2 (1.2)	2 (1.9)
	Current anti-endocrine therapy *		
		120 (71.9)	72 (70.6)
Treatment—Ovarian cancer	Surgery alone	4 (22.2)	2 (28.6)
ITT (*n* = 18)	Surgery + chemotherapy	14 (77.8)	5 (71.4)
PP (*n* = 7)			
	Current anti-endocrine therapy *	4 (22.2)	2 (28.6)
Treatment—Endometrial cancer	Surgery alone	11 (91.7)	6 (85.7)
ITT (*n* = 12)	Surgery + chemotherapy	1 (8.3)	1 (14.3)
PP (*n* = 7)			
Current status	No evidence of disease	191 (93.6)	112 (93.3)
	Recurrent/metastatic disease	13 (6.4)	8 (6.7)

* anti endocrine therapy refers to aromatase inhibitor or tamoxifen use with or without ovarian suppression.

**Table 2 cancers-16-01127-t002:** Characteristics of subjects with improvement in QoL > 5 from baseline to 6 months (vs. trivial or no improvement or no follow up).

Characteristic	Not Substantially Improved (*n* = 99)	Improvement ≥ 5 (*n* = 90)	Total (*n* = 189)	*p* Value
Baseline global health status				<0.001
Median (Q1, Q3)	67.0 (58.0, 83.0)	50.0 (42.0, 67.0)	67.0 (50.0,75.0)	
Started Sleepio				0.428
No	18 (18.2%)	12 (13.3%)	30 (15.9%)	
Yes	81 (81.8%)	78 (86.7%)	159 (84.1%)	
Completed Sleepio				0.008
No	76 (76.8%)	52 (57.8%)	128 (67.7%)	
Yes	23 (23.2%)	38 (42.2%)	61 (32.3%)	
Medications at six months				<0.001
Stopped	42 (42.4%)	17 (18.9%)	59 (31.2%)	
Not stopped	41 (41.4%)	60 (66.7%)	101 (53.4%)	
Unknown	16 (16.2%)	13 (14.4%)	29 (15.3%)	
Third level education				1
No	14 (14.1%)	13 (14.4%)	27 (14.3%)	
Yes	85 (85.9%)	77 (85.6%)	162 (85.7%)	
Never smoked				0.655
No	37 (37.4%)	37 (41.1%)	74 (39.2%)	
Yes	62 (62.6%)	53 (58.9%)	115 (60.8%)	
Units of alcohol				0.86
N-Miss	0	1	1	
Median (Q1, Q3)	2.0 (0.5, 5.0)	2.0 (0.0, 5.0)	2.0 (0.0, 5.0)	
Exercise ≥ 5 days/w				0.243
No	50 (50.5%)	37 (41.1%)	87 (46.0%)	
Yes	49 (49.5%)	53 (58.9%)	102 (54.0%)	
Breast cancer diagnosis				0.052
No	22 (22.2%)	10 (11.1%)	32 (16.9%)	
Yes	77 (77.8%)	80 (88.9%)	157 (83.1%)	
VMS same or worse at night				0.714
No	20 (20.2%)	16 (17.8%)	36 (19.0%)	
Yes	79 (79.8%)	74 (82.2%)	153 (81.0%)	
VMS worst at night				0.382
No	48 (48.5%)	50 (55.6%)	98 (51.9%)	
Yes	51 (51.5%)	40 (44.4%)	91 (48.1%)	
Cognitive symptoms				0.75
No	69 (69.7%)	65 (72.2%)	134 (70.9%)	
Yes	30 (30.3%)	25 (27.8%)	55 (29.1%)	
Psychological symptoms				0.869
No	72 (72.7%)	67 (74.4%)	139 (73.5%)	
Yes	27 (27.3%)	23 (25.6%)	50 (26.5%)	
Sexual symptoms				0.843
No	84 (84.8%)	75 (83.3%)	159 (84.1%)	
Yes	15 (15.2%)	15 (16.7%)	30 (15.9%)	
Musculoskeletal symptoms				0.882
N-Miss	0	1	1	
No	58 (58.6%)	54 (60.7%)	112 (59.6%)	
Yes	41 (41.4%)	35 (39.3%)	76 (40.4%)	
Sleep symptoms				0.366
No	40 (40.4%)	30 (33.3%)	70 (37.0%)	
Yes	59 (59.6%)	60 (66.7%)	119 (63.0%)	
Current anti endocrine therapy				0.655
No	40 (40.4%)	33 (36.7%)	73 (38.6%)	
Yes	59 (59.6%)	57 (63.3%)	116 (61.4%)	

**Table 3 cancers-16-01127-t003:** Frequency of gabapentin prescription in patients who accessed Sleepio.

	*n*	SCI at Baseline (SEM)	Prescribed Gabapentin (*n*)	SCI at 6 Months (SEM)	Continued Gabapentin (*n*)
Sleepio completers	62	7.5 (0.4)	50	17.4 (0.8)	41
At least 1 session of Sleepio	161	8.3 (0.4)	134	17.1 (0.5)	74
No Sleepio	43	9.5 (1.2)	35	18.4 (1.6)	11

## Data Availability

The raw data supporting the conclusions of this article will be made available by the authors on request.

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
