# Peer review of "Multimodal, Technology-Assisted Intervention for the Management of Menopause after Cancer Improves Cancer-Related Quality of Life—Results from the Menopause after Cancer (Mac) Study"

_cancers, 2024, doi:10.3390/cancers16061127_

Round 1
Reviewer 1 Report
Comments and Suggestions for Authors
This is a trial of a multimodal intervention for menopausal symptoms in women with a cancer history and contraindication for hormonal therapy. This topic is extremely important in cancer survivorship given the significant number of women who go into menopause younger than the general population due to treatments. The ability to offer effective interventions for women with contraindications to estrogen use is vital in this population, many of whom are breast cancer survivors. I would like to commend the authors for undertaking this trial and to thank them for this submission. The paper is well written and organized in an understandable format that was easy to follow. Conclusions are reasonable and based on a realistic interpretation of the data.
I do have a question. Can you please elaborate on the decision to pursue a single arm trial? As is mentioned in the discussion, because of the trial design it is hard to interpret which parts of the multimodal intervention are responsible for improvements. I assume this was to allow patients access to their best chance at experiencing improvement, but if you could please discuss how the study design was made further for the reader I think the reporting of your study would be more robust. One can imagine a "placebo" arm involving only the self management strategies or in combination with CBT.
Author Response
Thank you very much for your review of our work.
You are absolutely correct in that we chose the single arm design to give participants the best chance of experiencing improvement. In addition to this, the composite nature of the intervention it was difficult to predict effect sizes. Now that this study is complete, future studies could randomize participants to receive certain parts of the intervention as you suggested.
We have added these comments to the revised manuscript on line 380 to 389.
Best regards,
Fionán Donohoe
Reviewer 2 Report
Comments and Suggestions for Authors
Very well written and a very important and relevant topic.
In the discussion you mention that the largest improvement in QoL scores was at 4 weeks and then plateaus.
It would be useful to restate that in the conclusion.
Author Response
Thank you for your generous review of our work.
At your suggestion, we have restated that the most improvement was seen at 4 weeks and then maintained during the study in the conclusion. This can be seen on line 447 to 449.
Best regards,
Fionán Donohoe
Reviewer 3 Report
Comments and Suggestions for Authors
This is an interesting study. However, it should be emphasised that the mentioned drugs are “off-label”. Furthermore, there are some “editorial” mistakes you should correct. Subtitles don’t end with a full stop and are not complete sentences. Examples: Impact of composite MAC intervention on global health status. Baseline quality of life was the main predictor of benefit from MAC intervention.
Comments on the Quality of English LanguageSubtitles don’t end with a full stop and are not complete sentences. Examples: Impact of composite MAC intervention on global health status. Baseline quality of life was the main predictor of benefit from MAC intervention.
Author Response
Thank you for your review of our work.
We have taken your suggestions regarding the English language and adjusted the revised manuscript based on your review. All subtitles now end with full stops and are full sentences. We have also included that use of the medications in off label, as you suggested.
Best regards
Reviewer 4 Report
Comments and Suggestions for Authors
The Menopause after Cancer Study addresses a critical gap in addressing vasomotor symptoms in menopausal cancer patients, particularly those for whom conventional hormone therapy is not an option. By employing a multifaceted approach combining pharmacotherapy, digital cognitive behavioral therapy, self-management strategies, and social support, the study offers a comprehensive intervention tailored to the unique needs of this population. Strengths of the study include its robust methodology, large sample size, and clear assessment of both primary and secondary outcomes. The use of validated instruments to measure quality of life and symptom severity adds credibility to the findings. Moreover, the inclusion of a diverse range of cancer patients enhances the generalizability of the results.
One notable limitation is the lack of a control group (patients with natural menopause), which restricts the ability to establish causality definitively. While the observed improvements in QoL and symptomatology are promising, they could potentially be influenced by factors other than the intervention itself. Additionally, the relatively short follow-up period of 6 months may not capture long-term effects or sustainability of the intervention.
I think the valuable study could be improved by addressing the following points:
1) While the introduction mentions breast cancer as a common cancer type among participants, it could benefit from a more nuanced discussion of how different cancer types and treatments may influence menopausal symptoms.
2) Neither the introduction not the discussion does not address patient preferences or perspectives regarding the proposed interventions.
3) While the Discussion briefly mentions the higher-than-expected dropout rates and their potential association with the study duration, it could benefit from a more in-depth exploration of factors contributing to participant attrition.
Author Response
Thank you for your review of our work.
Based on your feedback, we have adjusted the introduction to highlight how cancer types other than breast cancer can be implicated in menopause symptoms.
We agree we did not discuss patient preferences regarding different aspects of the intervention. We did not collect this information during the course of the study but we agree this could have been a useful way of assessing participant preference for the different elements of the intervention and helped us to determine what elements were having the most impact.
In addition, we have added some more discussion on the reasons for participant attrition outside of the association with the duration of follow up. This can be seen on lines 401-409.
Best regards